# Beyond the Microscope: A Technological Overture for Cervical Cancer Detection

**DOI:** 10.3390/diagnostics13193079

**Published:** 2023-09-28

**Authors:** Yong-Moon Lee, Boreom Lee, Nam-Hoon Cho, Jae Hyun Park

**Affiliations:** 1Department of Pathology, College of Medicine, Dankook University, Cheonan 31116, Republic of Korea; vilimoon@hanmail.net; 2Department of Biomedical Science and Engineering (BMSE), Institute of Integrated Technology (IIT), Gwangju Institute of Science and Technology (GIST), Gwangju 61005, Republic of Korea; leebr@gist.ac.kr; 3Department of Pathology, Severance Hospital, College of Medicine, Yonsei University, Seoul 03722, Republic of Korea; CHO1988@yuhs.ac; 4Department of Surgery, Wonju Severance Christian Hospital, Wonju College of Medicine, Yonsei University, Wonju 26492, Republic of Korea

**Keywords:** AI-assisted diagnostics, PAP smear classification, cervical cancer screening, digital pathology, healthcare insurance

## Abstract

Cervical cancer is a common and preventable disease that poses a significant threat to women’s health and well-being. It is the fourth most prevalent cancer among women worldwide, with approximately 604,000 new cases and 342,000 deaths in 2020, according to the World Health Organization. Early detection and diagnosis of cervical cancer are crucial for reducing mortality and morbidity rates. The Papanicolaou smear test is a widely used screening method that involves the examination of cervical cells under a microscope to identify any abnormalities. However, this method is time-consuming, labor-intensive, subjective, and prone to human errors. Artificial intelligence techniques have emerged as a promising alternative to improve the accuracy and efficiency of Papanicolaou smear diagnosis. Artificial intelligence techniques can automatically analyze Papanicolaou smear images and classify them into normal or abnormal categories, as well as detect the severity and type of lesions. This paper provides a comprehensive review of the recent advances in artificial intelligence diagnostics of the Papanicolaou smear, focusing on the methods, datasets, performance metrics, and challenges. The paper also discusses the potential applications and future directions of artificial intelligence diagnostics of the Papanicolaou smear.

## 1. Introduction

Cervical cancer is the fourth most common cancer among women worldwide and is mainly caused by human papillomavirus (HPV) infection, with an estimated 604,000 new cases and 342,000 deaths in 2020 [1]. HPV is transmitted through sexual contact and can be prevented by vaccination and screening. Over 90% of new cases and deaths occur in low- and middle-income countries. Women infected with human immunodeficiency virus (HIV) are six times more likely to develop cervical cancer than women without HIV infection [2]. Cervical cancer can be cured if diagnosed early and treated promptly. Therefore, the World Health Organization (WHO) recommends HPV testing as the primary screening method, followed by treatment of pre-cancerous lesions or referral for further evaluation and management of invasive cancer, and WHO has adopted a global strategy to accelerate the elimination of cervical cancer, which involves reaching the 90-70-90 targets by 2030 [3]. These targets are 90% of girls vaccinated against HPV, 70% of women screened with a high-performance test, and 90% of women with cervical disease treated [3].

The Papanicolaou (PAP) smear is the most widely used screening method for cervical cancer. It involves collecting cells from the cervix and examining them under a microscope to identify any abnormalities (Figure 1). The PAP smear test can detect not only cervical cancer but also precancerous lesions that can be treated before they develop into cancer. The PAP smear test has been proven to be effective in reducing the mortality rate of cervical cancer by 70% [4]. However, the PAP smear test also has some limitations, such as:
It is time-consuming and labor-intensive, as it requires trained cytotechnologists or pathologists to manually review a large number of slides.It is subjective and inconsistent, as different experts may have different interpretations and opinions on the same slide.It is prone to human errors, such as misclassification, false negatives, false positives, or missed lesions.It has low sensitivity and specificity, as it may fail to detect some subtle or rare abnormalities or may confuse some benign conditions with malignant ones.

To overcome these limitations, AI techniques have been applied to PAP smear diagnosis in recent years. AI techniques can automatically analyze PAP smear images and classify them into normal or abnormal categories, as well as detect the severity and type of lesions. AI techniques can also provide quantitative and objective results that are consistent and reproducible. The incorporation of AI in cervical cell screening presents numerous potential advantages, such as heightened diagnostic precision, improved efficiency, and increased patient comfort. AI algorithms can process and analyze large datasets, identifying patterns and inconsistencies that might elude human detection. Furthermore, AI-facilitated screening could lessen the necessity for invasive procedures, reducing patient discomfort and enhancing outcomes.

It is paramount to seek innovative solutions that can augment the reliability and efficiency of cervical cancer screening, thereby improving patient outcomes and lessening healthcare burdens. The impetus for this review arises from the compelling need to refine cervical cancer screening methodologies, addressing the current techniques’ drawbacks and exploring AI’s potential to transform early detection. This paper aims to provide a comprehensive review of the recent advances in AI diagnostics of the PAP smear, focusing on the methods, datasets, performance metrics, and challenges. The paper also discusses the potential applications and future directions of AI diagnostics of the PAP smear.

## 2. Performance Metrics, Datasets from Image Patches to Whole Slide Images (WSIs), from Machine Learning (ML) to Vision Transformer

In this particular section, we will carry out an extensive comparative analysis of the performance metrics and datasets that are employed in AI diagnostics of the PAP smear in the literature reviewed (Table 1). Our objective is to offer a brief summary of the essential aspects. Detecting precancerous lesions for cervical cancer in a timely manner is crucial for effective treatment and prevention. Thus, it is imperative to investigate and scrutinize the diverse techniques and methods used for PAP smear diagnostics to enhance the precision and effectiveness of detection.

In 1992, the PAPNET, the first commercial automatic screening system, was approved. However, it was only authorized as a method of re-screening for slides that were initially deemed negative by cytologists [5]. The ThinPrep^®^ imaging system (Version 1.0, Hologic, Marlborough, UK), which was approved as a commercial screening product in 2004, employs a proprietary algorithm to choose the 22 most concerning fields of view (FOV). This has reduced the workload of pathologists while also increasing the accuracy of the process [6].

The ThinPrep^®^ imaging system uses liquid-based cytology, which has several advantages over traditional methods. For instance, it is viable to obtain a more representative exemplar of the cervix, which curtails the number of mistaken negatives. The system also eliminates the need for manual fixation and staining, which can introduce variability and artifacts.

Recent studies have shown that the ThinPrep^®^ imaging system is more sensitive than traditional cytology screening. Moreover, it is additionally more duplicable, which boosts its correctness. The software is proficient in identifying subtle modifications in cells that could be a sign of cancer, enhancing its effectiveness.

Although the ThinPrep^®^ imaging system has advantages, it also has limitations. Its implementation can be expensive and specialized training is required for operators. Additionally, certain variables, such as the presence of hemoglobin or mucus, can interfere with the investigation. In the year 2008, the emergence of the FocalPoint GS imaging system marked a significant milestone in the field of cervical cytology. The system was designed to identify 10 FOV of cervical cells most likely to be abnormal, which allowed for the stratification of risk and improvement of efficiency [7]. Although there are advantages to this automated system, certain assessments indicate that its cost-effectiveness is confined and may not be suitable for use in developing countries with low to medium development [8]. Additionally, the technology has weaknesses and depends on the final manual screening process [9]. Thus, analysts continue to examine the implementation of AI technology in cervical cytology, with a focus on improving efficiency. The integration of AI technology would enable the automation of the screening process, thereby alleviating the burden on cytopathologists and elevating the accuracy and efficiency of the outcomes. Furthermore, the adoption of AI would bring about heightened cost-effectiveness throughout the entire procedure, which would be particularly beneficial for countries with limited financial resources, where affordability remains a significant concern. By embracing AI technology, the screening process would undergo a remarkable transformation, characterized by swiftness, precision, and cost-efficiency, thereby widening its accessibility to a larger population.

Chankong et al. utilized fuzzy c-means clustering technology to segment single-cell images into the nucleus, cytoplasm, and background, thereby achieving whole-cell segmentation [8].

An investigation explored a segmentation model that utilizes images extracted from a PAP smear slide. The model utilized nucleus localization to differentiate normal and abnormal cells, combined with single-cell classification algorithms. The segmentation model achieved a high level of accuracy and sensitivity, respectively, with 91.7% consisting of Mask-RCNN [10].

Recent years have witnessed a transformation in the methods employed for classification, with the majority of approaches no longer depending solely on texture feature extraction or segmentation. One such novel approach involves segmenting cervical single-cell images into the nucleus, cytoplasm, and background, and then extracting morphological features to enable automatic multi-label classification. The consequences of this strategy have been exceedingly promising, with a precision rate above 93%, suggesting the potential efficacy of this methodology in automated sorting [11].

Another innovative approach involves the extraction of seven groups of texture features of cervical cells for classification, with the support vector machine (SVM) classifier demonstrating the highest accuracy and best performance. This strategy is extremely efficient at classifying the pictures with a high level of precision. However, it is worth noting that the precision percentage of the incorporated categorizer was solely 50% at the stain plane and 60% at the unit plane, which indicates the need for further refinement and optimization [12].

Researchers are finding automated categorization methods that do not rely on an accurate segmentation algorithm to be more and more attractive. One of these methods utilizes deep learning (DL) and transfer learning to classify cervical cells. The likelihood of achieving such exceptional performance through manual extraction of deep-level features from cell images for classification, with an accuracy of 98.3%, an AUC of 0.99, and a specificity of 98.3%, is low. This highlights the significant potential of utilizing advanced machine learning techniques, such as deep learning, for improving the accuracy and efficiency of cell image analysis in various medical applications [13].

The complexity and specificity of the task at hand, as described earlier, necessitated the use of a graph convolution network for the precise classification of cervical cells. This advanced machine learning technique achieved impressive results, with precision, sensitivity, specificity, and F-measure rates of 98.37%, 99.80%, 99.60%, and 99.80%, respectively. These findings demonstrate the potential of utilizing graph convolution networks for accurate and efficient analysis of complex medical images [14].

The examination directed by Bao et al. comprises a possible companion examination of a broad populace of females, involving 700,000 people who were going through screening for cervical carcinoma. The AI-assisted cytological diagnostic system employed in the study was validated, resulting in a total coincidence rate of 94.7%. Moreover, this synchronized with a marked upsurge in sensitivity of 5.8% (3.0% to 8.6%) in comparison to manual check. This study demonstrates that the integration of AI-assisted cytological examination can significantly improve the detection and classification of cervical cells and should be considered as a potential tool for guiding triage in cervical cancer screening programs [15].

Zhu and colleagues developed an AIATBS diagnostic system that utilized ThinPrep^®^ and artificial intelligence and showed higher sensitivity than the diagnosis conducted by experienced cytologists. In fact, the AIATBS system had a remarkable sensitivity of 94.74% when detecting CIN. These discoveries have noteworthy ramifications for the domain of cervical ailment diagnosis, as they propose that AI has the potential to significantly enhance the accuracy and sensitivity of existing diagnostic techniques [16].

Chen and his colleagues carried out research on CytoBrain, a screening system for cervical cancer that employs artificial intelligence. This system utilizes deep learning technology and comprises cervical cell segmentation, classification, and human-aided diagnosis visualization modules. The study mainly focuses on cell segmentation and classification components and proposes a compact VGG network called CompactVGG as the classifier. The researchers introduced a large dataset of 198,952 cervical cell images from 2312 participants, which were categorized into positive (abnormal), negative (normal), and junk categories. The CompactVGG structure features 10 convolutional layers, 4 max pooling layers, and 2 fully connected layers, totaling 1,128,387 parameters. The independent test group found evidence of CompactVGG’s accuracy being 88.30%, sensitivity being 92.83%, specificity being 91.03%, precision being 82.26%, and F1-score being 87.04%. These outcomes surpassed the Inception v3, ResNet50, and DenseNet121 models on all metrics. Furthermore, CompactVGG demonstrated superior time and classification performance compared with existing VGG networks on the Herlev and SIPaKMeD public datasets. In conclusion, the CytoBrain system with its CompactVGG classifier has the potential to improve cervical cancer screening through its accurate and efficient performance [17].

Wei and colleagues introduced an innovative module called InCNet, which enhances the multi-scale connectivity of the network while maintaining efficiency. This module is seamlessly integrated into a lightweight model named YOLCO (You Only Look Cytopathology Once) and is specifically designed to extract features from individual cells or clusters. To evaluate their approach, the authors curated a novel dataset comprising 2019 whole slide images (WSIs) obtained from four different scanners. The dataset includes annotations for both normal and abnormal cells and is publicly accessible for research purposes.

In order to assess the performance of their method, the authors compare it with a conventional model that employs a ResNet classifier. The evaluation was conducted on 500 test WSIs. The results demonstrate that the proposed method outperforms the conventional model across most metrics, except for specificity, where the conventional model exhibits a slight advantage. The AUC score achieved by the new method is 0.872, while the conventional model obtains a score of 0.845. Moreover, the accuracy of the proposed method reaches 0.836, surpassing the accuracy of 0.824 achieved by the conventional model.

Furthermore, the authors showcase the clinical relevance of their method by illustrating its ability to detect sparse and minute lesion cells in cervical slides. This capability is particularly challenging for human experts and conventional models. The authors assert that their method has the potential to enhance the diagnosis and screening of cervical cancer, thereby contributing to improved healthcare outcomes [18].

Cheng and colleagues developed an innovative system for cervical cancer screening that utilizes deep learning techniques on WSIs. This computer-aided approach has the potential to significantly improve the accuracy and efficiency of cervical cancer screening, offering a promising new avenue for early detection and treatment. The system consists of three models, namely, low-resolution lesion localization, high-resolution cell identification/ranking, and recurrent neural network WSI classification models. To evaluate the efficacy of their system, they used 3545 WSIs from five hospitals and scanners, with 79,911 annotations. In independent testing conducted on 1170 WSIs, the system demonstrated 93.5% specificity and 95.1% sensitivity, similar to three experienced cytopathologists. The system also identified the top 10 lesion cells with a true positive rate of 88.5% on 447 positive slides, surpassing the Hologic ThinPrep^®^ Imaging System. The computational efficiency of the system is remarkable, given its ability to process giga-pixel WSIs in around 1.5 min per graphic processing unit (GPU), which illustrates its effectiveness in actual screening scenarios [19].

Wang and colleagues proposed an innovative approach for detecting cervical high-grade squamous intraepithelial lesions and squamous cell carcinoma screening in PAP smear WSIs using cascaded fully convolutional networks. This ingenious method involves a step-by-step process of utilizing fully convolutional networks to accurately identify and classify abnormalities in WSIs, which could have significant implications for improving cervical cancer screening and diagnosis. Their investigation, released in a top-tier medical journal, evaluates their proposed deep learning screening system’s efficacy against other state-of-the-art approaches like U-Net, SegNet, and a previous method. The creators shared that their suggested technique obtained a precision of 0.93, recall of 0.90, F-measure of 0.88, and Jaccard index of 0.84 on a 143-WSI dataset, demonstrating remarkable performance over the other methods. Additionally, the proposed method demonstrated a remarkable processing speed of 210 s per WSI, which is 20 times and 19 times faster than U-Net and SegNet, respectively. The potential for an AI system to produce these exceptional results and suggest a method for quickly and accurately identifying severe cervical pathologies in real-world clinical environments could be highly advantageous for the medical community, particularly in areas with limited resources. The results of this analysis demand additional inquiry and authentication in larger and more varied datasets [20].

Kanavati and team designed a model for detecting cervical cancer in liquid cytology WSIs that utilized deep learning, consisting of trained convolutional and recurrent neural networks. The model was tested on 1605 training and 1468 multi-test set WSIs and achieved an ROC AUC range of 0.89–0.96, indicating its potential to assist in screening. Furthermore, the model generates neoplastic cell probability heatmaps that help identify suspicious regions. The model exhibited either comparable or superior accuracy, sensitivity, and specificity when compared with semi-automated devices. Therefore, it has the potential to standardize screening and reduce fatigue. The results of this study suggest that incorporating deep learning into cervical cancer screening could have a substantial impact on the accuracy and efficiency of the screening process. Furthermore, the integration of this model into healthcare may lead to the premature detection of cervical cancer and potentially rescue lives [21].

Hamdi and colleagues developed a novel approach for the analysis of whole slide cervical images and cancer staging using a hybrid deep learning system to generate a combination of models that includes ResNet50, VGG19, GoogLeNet, Random Forest, and support vector machine. The team also utilized the Active Contour Algorithm for segmentation and fused deep model features as another approach. The ResNet50-VGG19-Random Forest model achieved outstanding results on a dataset of 962 cervical squamous cell images. Particularly, the prototype accomplished 97.4% sensitiveness, 99% exactness, 99.6% exactitude, 99.2% selectivity, and 98.75% AUC, which displays noteworthy potential for beforehand detection. Considering the encouraging outcomes of this research, it is probable that upcoming studies may comprise clinical data and more comprehensive, diverse datasets, as well as an investigation of further deep learning models. Overall, this proposed plan of action has the potential to advance the field of cervical cancer diagnosis and improve patient outcomes [22].

Diniz and colleagues conducted a study that involved comparing ten deep convolutional neural networks for classifying cervical cells in PAP smears into two, three, and six categories using conventional cytology images. They proposed an ensemble of the top three models and used extensive data augmentation and balancing. The appraisal of the investigation utilized accuracy, specificity, F1-score, cross-validation, recall, and precision. The ensemble model outperformed individual and prior architectures across all classification tasks. The recall achieved for the two, three, and six classes were 0.96, 0.94, and 0.85. The study highlights the potential of ensemble deep learning in improving the accuracy of cervical cancer screening and, subsequently, patient outcomes. The findings of the investigation suggest that the utilization of an assembly model may result in improved efficacy when compared with individual models. The authors further established the efficacy of enhancing data through augmentation and balancing to enhance the precision of the model. This study provides valuable insights into the use of advanced ensemble deep learning in cervical cancer screening and has the potential to inspire further research in this field [23].

Tripathi and colleagues conducted a study that involved the classification of five cervical cancer cell types in 966 PAP smear images using four pre-trained deep models. These architectures were constructed by a team of highly skilled engineers and were carefully tested to ensure optimal performance. ResNet-152 reached the maximum precision of 94.89%, with VGG-19, ResNet-50, and VGG-16 close on its heels. Furthermore, the study reported class-wise performance, and certain combinations of models achieved 100% recall and precision for specific classes. The investigation accentuates the potential of deep transfer learning for precise classification and implies that further progressions in original models, hyperparameter optimization, and clinical data integration could boost the accuracy even further. In line with the data, this analysis provides valuable insights for future research in the field of cervical cancer grouping. These findings can contribute to improving cervical cancer diagnosis and patient outcomes [24].

Zhou and colleagues presented a comprehensive cervical screening framework that includes three stages: cell detection, image classification, and case classification. The first stage involved detecting cells using the RetinaNet model, which achieved an impressive 0.715 average precision in just 0.128 s per image. The following step integrated an innovative patch encoder-fusion component for image classification, achieving a 0.921 accuracy and 0.903 sensitivity. For the final phase, a support vector machine functioned as a case classifier, providing an accuracy of 0.905 and sensitivity of 0.891, outclassing other models. These numerical results clearly demonstrate the framework’s effectiveness in leveraging cell cues to improve the robustness of case diagnosis. In general, this inventive system shows vast potential for enhancing the detection and diagnosis of cervical cancer, ultimately resulting in improved health outcomes for women. Additional exploration is warranted to affirm the framework’s effectiveness in larger and more diverse patient populations [25].

The CervixFormer proposal has demonstrated a considerable level of efficacy in the classification of PAP smear whole slide images. Unlike other inferior transformer and convolutional models, this particular model has exhibited commendable performance on both private six-class and public four-class datasets. Additionally, the program has showcased strong binary, three-class, and five-class cellular classification precision, recall, accuracy, and F1-scores. The CervixFormer proposal has utilized data augmentation and stain normalization techniques to enhance diversity and staining invariance across the datasets. The incorporation of Swin Transformer subnetworks into the model has facilitated multi-scale feature learning through the fuzzy rank fusion approach. Consequently, the GradCAM visualizations on the important regions have provided clinical interpretability of the model’s outputs. Overall, the CervixFormer proposal has shown promise as a scalable and reliable solution for cervical screening and diagnosis, with the potential for clinical deployment [26].

In brief, the available data suggest that AI exhibits remarkable detection rates and precision when it comes to cytology. Nonetheless, there remains the potential for conducting extensive research and delving into innovative applications within this field. For instance, the development of AI microscopes can potentially revolutionize cytology screening by enhancing its efficiency and accuracy. Additionally, an AI assistant colposcopy represents a highly advanced instrument that can assist in the identification and evaluation of cervical cancer [27]. This particular innovation possesses the capacity to transform the realm of cervical cancer treatment, as it has the ability to furnish healthcare practitioners with invaluable perspectives and assistance. Various potential paths regarding the treatment of cervical cancer exist, highlighting immunotherapy, targeted therapy, and the use of PARP inhibitors [28].

**Table 1 diagnostics-13-03079-t001:** Summary of remarkable works focusing on AI-assisted PAP smear (* Please see footnote for abbreviation and Appendix A for details).

References	Year	Datasets (Number of Images)	Metrics	Methods
Chankong et al. [8]	2014	ERUDIT (552) Herlev (917)	Accuracy 93.78 to 99.27%	Bayesian classifier * + KNN * + ANN *
Wang et al. [11]	2019	Private (362)	Sensitivity 94.25%; specificity 93.45%	Mean-Shift clustering algorithm *
Zhang et al. [13]	2017	Herlev (917) HEMLBC (2370)	Accuracy 98.30 to 98.6%; specificity 98.30 to 99.00%	CNN * + transfer learning *
Shi J et al. [14]	2021	SIPAKMeD (4049)	Accuracy 98.37%; sensitivity 99.80%	CGN *
Bao et al. [15]	2020	Cervical cancer screening program (703,103)	CIN1+ Sensitivity 88.9%; specificity 95.8%; CIN2+ Sensitivity 90.10%; specificity 94.80%	DL *
Zhu et al. [16]	2021	Cytological image biopsy diagnosis proven (980)	Sensitivity 94.74%	AIATBS *
Chen et al. [17]	2021	WSI (198,952)	Accuracy 88.30%; sensitivity 92.83%; specificity 91.03%; precision 82.26%; F1-score 87.04%	CompactVGG *
Wei et al. [18]	2021	WSI (2019)	Accuracy 80.80%; sensitivity 90.60%; specificity 71.00%	YOLCO *
Cheng et al. [19]	2021	WSI (3545)	Sensitivity 93.50%; specificity 95.10%	RNN *
Wang et al. [20]	2021	WSI (143)	Precision 93.00%; recall 90.00%; F1-score 88.00%	FCN *
Kanavati et al. [21]	2022	WSI (1605)	Accuracy 90.00%; sensitivity 86.00%; specificity 91.00%	CNN + RNN
Hamdi et al. [22]	2023	WSI (962)	Accuracy 99.00%; sensitivity 97.40%; specificity 99.20%; precision 99.60%	RF * + ResNet50 * + VGG19 *
Diniz et al. [23]	2021	CRIC (3233)	Accuracy 96.00%; recall 94.00%; specificity 97.00%; precision 94.00%; F1-score 94.00%	MobileNet * + InceptionNet * + EfficientNet *
Tripathi et al. [24]	2021	SIPAKMED (966)	Accuracy 94.89%	ResNet-152 *
Zhou et al. [25]	2021	WSI (237)	Accuracy 90.50%; sensitivity 89.10%; F1-score 86.70%	SVM * + RetinaNet * + Encoder *
	2023	Mendeley (963) SIPaKMeD (4049) Dankook University Hospital (100,000) AI-Hub (20,000)	Accuracy 95.00%; recall 95.00%; precision 97.00%; F1-score 95.00%	GRAD-CAM * + Swin Transformer *
Khan et al. [26]				

* KNN: K-nearest neighbor; ANN: artificial neural network; CNN: convolutional neural network; CGN: convolutional graph network; DL: deep learning; AIATBS: Artificial Intelligence-Assisted ThinPrep^®^ Imaging System; YOLCO: You Only Look Cytopathology Once; RNN: recurrent neural network; FCN: fully convolutional network; RF: Random Forest; SVM: support vector machine; GRAD-CAM: Gradient-weighted Class Activation Mapping.

## 3. Discussion

The detection and treatment of cervical cancer, a key area of women’s healthcare, can be transformed by the implementation of AI. The current manual screening methods, such as the PAP smear test, are limited by the complexity, monotony, and subjectivity involved in the human examination of cytology slides, leading to inefficiency and inaccuracy. To overcome these limitations, computer-aided diagnosis leverages advanced algorithms to analyze cell morphology and classify smears quickly. AI can enhance the accuracy and specificity of screening and diagnostic programs, overcome time constraints, and prevent bias caused by subjective factors. By allowing cervical cancer screening to be executed in areas with limited resources, artificial intelligence has the potential to significantly decrease the prevalence of cervical cancer.

The employment of human intelligence suggests multiple challenges that require attention to adequately integrate artificial intelligence (AI) algorithms. A major obstacle is the insufficiency of information, which often requires millions of data points for AI to achieve satisfactory efficacy levels. Unfortunately, current clinical data may lack markers, have uncertain quality, and be scarce, making it difficult to manage medical data effectively. It is critical to establish standardized and extensive databases in the future while considering data security concerns and over-fitting, which can lead to over-diagnosis. Clinical practice consistently demonstrates the reliability of AI-based models. However, it is noteworthy that AI is not designed to supplant clinicians but rather to function as a supportive diagnostic tool. Furthermore, AI may result in system paralysis, necessitating technical skills for maintenance and specialized training, and systems must be implemented to ensure proper maintenance.

Recent developments in the field of deep learning have generated a renewed sense of optimism with regards to achieving dependable automated screening. Unlike human observers, AI algorithms have the ability to analyze large datasets and identify subtle patterns or anomalies, resulting in more accurate diagnoses or outcomes. Additionally, AI has the potential to improve patient comfort and well-being by utilizing less invasive procedures for detecting cancer cells from medical images. These advantages highlight the significant potential of AI in the field of medical imaging and diagnosis, and underscore the importance of continued research and development in this area. The amalgamation of AI into cervical screening offers an array of advantages like increased diagnostic precision, efficiency, enhanced patient comfort, decreased healthcare expenses, and superior outcomes. According to a cost-effectiveness analysis by Chen et al., AI-based PAP smear screening can save up to 30% of healthcare costs compared with conventional cytology screening [17].

Nevertheless, the identification of these prospects is met with impediments such as ensuring the precision of data, obtaining regulatory approval, addressing ethical considerations, clinically validating the functionality of systems, and encouraging patient acceptance. Moreover, obstructions persist, encompassing inadequate labeled data, variations in staining, the opacity of the model, and the requisite compliance with regulatory and ethical standards. To corroborate the efficacy of artificial intelligence as a screening adjunct, further extensive and comprehensive exploration in genuine, real-life scenarios is imperative.

The field of AI-assisted PAP smear classification research has a lengthy history in comparison to other domains, and its progress has nearly plateaued due to the accumulation of findings. Nonetheless, AI-based methods have not gained widespread use in clinical practice, and we contend that three obstacles must be surmounted. Firstly, the accurate classification of PAP smears necessitates the analysis of WSIs, which, given that diagnostic patches compose only a small proportion of tens of thousands of image patches in a slide, demands high-throughput and precise analysis of thousands of patches from WSIs. Secondly, the scanning of digital slides, which necessitates high-performance and high-cost devices, is required for WSI analysis, and their global penetration remains limited due to pricing. Finally, the absence of medical insurance support to subsidize the costs of expensive digital scanners required to address the second challenge and the high-end GPU necessary to tackle the first challenge present daunting obstacles to clinical adoption. In short, AI-based PAP smear screening is the processing of large-scale and high-resolution WSIs, which require high-performance and high-cost devices and GPUs. To overcome this challenge, we used a novel image analysis method based on object segmentation and CNN that can efficiently segment and quantify cells in WSIs without compromising accuracy.

In brief, the technical performance of AI-based techniques is commendable; however, there are still economic and infrastructure challenges that need to be addressed for these techniques to have a widespread impact in domains such as PAP smear screening. Artificial intelligence may have applications beyond early screening and diagnosis, including the prediction of prognosis, prevention, and treatment of cervical cancer. Additional inquiry is required to be performed on the subject of therapy and prognosis for the purpose of efficient therapy selection and to enable the global elimination of cervical cancer. As the incidence of cervical adenocarcinoma and other rare pathological types increases, AI should be utilized for early diagnosis in the future. Furthermore, AI can be utilized for the noninvasive differentiation of cervical cancer from other diseases. With the additional advancement of AI technologies, the estimation of cervical cancer can be significantly improved, leading to noteworthy enhancements in cervical cancer screening and diagnosis, refinement of staging systems, and better patient prognosis.

## 4. Conclusions

The execution of AI in the screening of cervical cancer provides a hopeful alteration in the detection and management of this ailment. The current screening procedures, such as PAP smears, possess insufficiencies with regard to their effectiveness and precision. PAP smears require manual examination, which can be intricate, tedious, and subject to human subjectivity and fatigue. Nevertheless, the utilization of computer-aided diagnosis strives to surmount these limitations by implementing sophisticated algorithms that can swiftly evaluate cell morphology and classify slides on the basis of the probability of abnormality or malignancy. With recent advancements in deep learning, there is newfound hope in achieving the long-awaited goal of dependable automated screening. The utilization of AI algorithms is capable of analyzing tremendous datasets and detecting patterns and anomalies that may go unnoticed by human observers. Undoubtedly, this technological innovation has the potential to significantly improve patient outcomes by enabling more precise diagnoses and treatments. As AI and other advanced machine learning techniques continue to evolve and improve, they offer promising new avenues for enhancing the accuracy and efficiency of medical imaging and diagnosis, ultimately leading to better patient care and outcomes.

Additionally, the utilization of AI screening methods holds the possibility of mitigating the need for invasive procedures, thereby reducing patient discomfort and enhancing patient results. Also, AI-driven PAP smear diagnostics may serve as a telemedicine and mobile health system, simplifying remote and accessible screening and diagnosis of cervical cancer. Moreover, AI-based PAP smear diagnostics can function as a research and education system that can foster the dissemination and advancement of knowledge and skills in cervical cancer. The implementation of AI for the purpose of identifying cervical cancer is hindered by various obstacles and challenges, such as deficient data quality, lack of regulatory authorization, ethical concerns, clinical validation, and patient skepticism despite its vast potential for significant gains. The potential of AI in revolutionizing cervical cancer screening programs worldwide through the efficient use of digitized cytology is considerable, but it will require careful consideration and effective action to overcome the challenges and limitations.

Effective integration of workflow and ongoing model refinement is essential to fully capitalize on the benefits of AI while minimizing risks. Although there are still challenges, evidence indicates that AI could revolutionize the detection of cervical cancer and other illnesses in the upcoming years. However, AI-assisted cytologic evaluation should not replace human governance and accountability. Rather, it should function as a tool to enhance human abilities and enhance patient outcomes.

## Figures and Tables

**Figure 1 diagnostics-13-03079-f001:**
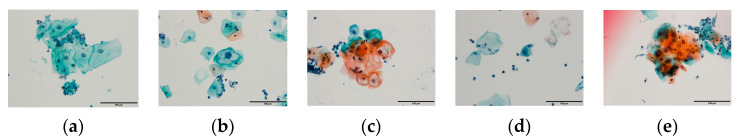
The PAP smear cytology of uterine cervical cells classified by disease progression: (**a**) normal; (**b**) atypical squamous cell of undetermined significance (ASCUS); (**c**) low-grade squamous intraepithelial lesion (LSIL); (**d**) atypical squamous cell cannot exclude HSIL (ASC-H); and (**e**) high-grade squamous intraepithelial lesion (HSIL) (segmented and labeled by the authors).

## Data Availability

Some or all datasets generated during and/or analyzed during the current study are not publicly available but are available from the corresponding author upon reasonable request.

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
