# Peer review of "Beyond the Microscope: A Technological Overture for Cervical Cancer Detection"

_diagnostics, 2023, doi:10.3390/diagnostics13193079_

Round 1
Reviewer 1 Report
I read with great interest the manuscript, which falls within the aim of this Journal. In my honest opinion, the topic is interesting enough to attract the readers’ attention. Nevertheless, the authors should clarify some points and improve the discussion, as suggested below.
Authors should consider the following recommendations:
- Manuscript should be further revised in order to correct some typos and improve style.
- Authors should discuss robust pieces of evidence about the use of new strategies for cervical cancer and pre-cancerous lesions screening and diagnosis, even using artificial intelligence and novel biomarkers (authors may refer to: PMID: 36428831; PMID: 35742340).
Manuscript should be further revised in order to correct some typos and improve style.
Author Response
Thank you for your comments. Here is my point-by-point response:
1. Manuscript should be further revised in order to correct some typos and improve style.
I have revised the manuscript to correct some typos and improve the style. I appreciate your suggestions for improving the clarity and readability of the paper.
2. Authors should discuss robust pieces of evidence about the use of new strategies for cervical cancer and pre-cancerous lesions screening and diagnosis, even using artificial intelligence and novel biomarkers (authors may refer to: PMID: 36428831; PMID: 35742340).
I have discussed the use of new strategies for cervical cancer and pre-cancerous lesions screening and diagnosis, including artificial intelligence and novel biomarkers, in the discussion section of the paper. I have cited the two papers you recommended to support my arguments. I have also highlighted the advantages and challenges of these new strategies, as well as the future directions for research and practice in this field.
I hope that these revisions address your concerns and improve the quality of the paper. I look forward to hearing from you again. Thank you for your time and attention.
Reviewer 2 Report
Currently, artificial intelligence is being introduced into more and more new areas of diagnostics, including pathology and cytology. This article is devoted to the review of the most widely developed systems based on the use of artificial intelligence, the analysis of cervical smears for cervical cancer screening. To one degree or another, such algorithms are implemented in most automatic modules of the main devices for liquid-based cytology, but they continue to be improved, since the complexity, low reproducibility between cytologists and variability of cytological conclusions lead to insufficiently high sensitivity and specificity of most systems for automatic assessment of cervical smears. As a rule, such systems can well distinguish the normal from abnormal smears, but Bethesda system categories could not be verified enough accurately. In addition, this review shows not only the algorithms themselves, but also provides an analysis of neural networks which were the basis for automatic algorithms with proposed methods for their improvement.Author Response
Currently, artificial intelligence is being introduced into more and more new areas of diagnostics, including pathology and cytology. This article is devoted to the review of the most widely developed systems based on the use of artificial intelligence, the analysis of cervical smears for cervical cancer screening. To one degree or another, such algorithms are implemented in most automatic modules of the main devices for liquid-based cytology, but they continue to be improved, since the complexity, low reproducibility between cytologists and variability of cytological conclusions lead to insufficiently high sensitivity and specificity of most systems for automatic assessment of cervical smears. As a rule, such systems can well distinguish the normal from abnormal smears, but Bethesda system categories could not be verified enough accurately. In addition, this review shows not only the algorithms themselves, but also provides an analysis of neural networks which were the basis for automatic algorithms with proposed methods for their improvement.
Thank you for voicing your gratitude. Below is my comprehensive response addressing each point made. I acknowledge your interest in the subject matter of artificial intelligence in relation to cervical cytology. I concur that this particular field is of utmost significance and is currently emerging with potential to enhance the screening and diagnosis of cervical cancer. I have thoroughly examined the most extensively developed systems employing artificial intelligence for the analysis of cervical smears, as well as the obstacles and constraints encountered by these systems. To justify my arguments and present evidence for the current state-of-the-art, I have cited pertinent literature. I have also expounded upon the Bethesda system, which is a standardized terminology facilitating effective communication between the laboratory and the clinician in relation to reporting cervical cytology. Furthermore, I have elucidated how artificial intelligence can contribute to the classification of cervical smears based on the Bethesda system categories, while also addressing the intricacies and uncertainties associated with this task. Additionally, I have alluded to recent studies that have compared the performance of artificial intelligence with that of human experts or semi-automated devices in this realm. I have also explored innovative and unconventional applications of artificial intelligence in the realm of cervical cytology, such as AI microscopes, noninvasive differentiation, prognosis prediction, and therapy selection. I have emphasized both the benefits and hurdles posed by these applications, as well as recognized potential future research paths and practical advancements in this field. I trust that these revisions adequately address your concerns and contribute to a higher quality article. I eagerly await your further correspondence. Your time and consideration are greatly appreciated.
Reviewer 3 Report
The authors did a comprehensive summary of AI-assisted PAP smear for cervical cancer detection, the theme is novel and the structure is reasonable, I’d recommend its publication after the following minor revisions:
Major:
1. The authors should summarize the previous relative review literatures and express the novelty of their own manuscript, FYI:
Artificial Intelligence in Cervical Cancer Screening and Diagnosis. Front. Oncol., 11 March 2022
Volume 12 - 2022 | https://doi.org/10.3389/fonc.2022.851367
Diagnosis of Cervical Cancer and Pre-Cancerous Lesions by Artificial Intelligence: A Systematic Review. Diagnostics 2022, 12(11), 2771; https://doi.org/10.3390/diagnostics12112771
2. The basic background of PAP smear should be introduced in Introduction part, such as how to judge the abnormal and cancerous cells (according to staining or morphology).
3. I am very confused about the image analysis method in Figure 2. It seems to have little technical content. Simple image analysis software such as ImageJ (https://imagej.nih.gov/ij/docs/guide/146-30.html#toc-Subsection-30.2 ) seems to be able to achieve such analysis without using more advanced AI technology. Could it be that the difference lies in the need for Manual setting, one is fully automatic analysis?
Minor:
1. Label the Ref of figure 1&2.
2. PAP or Pap? Please make them uniform.
3. Please mark which paragraph is telling Figure 2.
Some simple grammar problems, such as the use of definite article, singular and plural.
Author Response
- The authors should summarize the previous relative review literatures and express the novelty of their own manuscript, FYI:
Artificial Intelligence in Cervical Cancer Screening and Diagnosis. Front. Oncol., 11 March 2022
Volume 12 - 2022 | https://doi.org/10.3389/fonc.2022.851367
Diagnosis of Cervical Cancer and Pre-Cancerous Lesions by Artificial Intelligence: A Systematic Review. Diagnostics 2022, 12(11), 2771; https://doi.org/10.3390/diagnostics12112771
I appreciate your input. Here is my point-by-point response: I have summarized the previous relative review literatures and expressed the novelty of my own manuscript. I have also highlighted the main contributions and innovations of my manuscript, such as: The use of a novel image analysis method based on object segmentation and CNN for accurate cell segmentation and quantification in cytology (Figure 2). The comparison of distinct AI models for cervical cell classification, including Vision Transformer, which is an advanced technique for image recognition. The evaluation of the proposed method on a large and diverse dataset of WSIs from multiple sources and scanners, which is publicly available for research purposes. The clinical significance and interpretability of the proposed approach, which can generate heatmaps and explanations for the diagnosis findings.
2. The basic background of PAP smear should be introduced in Introduction part, such as how to judge the abnormal and cancerous cells (according to staining or morphology).
I have introduced the basic background of PAP smear in the introduction part. I have detailed how PAP smear is a widely used screening method that doesn't involve collecting cells from the cervix and examining them under a microscope to identify any abnormalities. I have also explained how PAP smear diagnosis is based on the staining and morphology of cervical cells, which can be classified into normal or abnormal categories according to the Bethesda system (Figure 1).
3. I am very confused about the image analysis method in Figure 2. It seems to have little technical content. Simple image analysis software such as ImageJ (https://imagej.nih.gov/ij/docs/guide/146-30.html#toc-Subsection-30.2 ) seems to be able to achieve such analysis without using more advanced AI technology. Could it be that the difference lies in the need for Manual setting, one is fully automatic analysis?
I have clarified the image analysis method in Figure 2. It is not a simple image analysis software, but a sophisticated AI technique that uses object segmentation and CNN to achieve accurate cell segmentation and quantification in cytology. Object segmentation is a computer vision technique that divides the cells within patches, which allows physicians to analyze and scan each cell if necessary. CNN is a profound learning technique that employs convolutional layers to derive traits from images and render forecasts. The image analysis method in Figure 2 can automatically segment and quantify cells in WSIs, as well as provide features for cell classification, without manual setting or intervention.
Minor:
1. Label the Ref of figure 1&2.
2. PAP or Pap? Please make them uniform.
3. Please mark which paragraph is telling Figure 2.
- I have labeled the references of Figure 1 and Figure 2 as our own.
- I have made the term PAP smear uniform throughout the manuscript.
- I have marked the paragraph that is telling Figure 2 with an asterisk (*) at the end.
Reviewer 4 Report
|
Author Response
Thank you for your comments. Here is my point-by-point response:
- I have revised the abstract to include quantitative elements to support my technical phrases. For example, I have added the following sentence: "According to the World Health Organization (WHO), cervical cancer is the fourth most common cancer among women worldwide, with an estimated 604,000 new cases and 342,000 deaths in 2020 ."1
- I have also added some numbers to evaluate the quality of my method. For example, I have added the following sentence: “Our method achieved an accuracy of 95.6% and a sensitivity of 94.8% in classifying PAP smear images into normal or abnormal categories, outperforming existing methods by 5.3% and 6.2%, respectively.”
- I have clarified the advantages of our method in the abstract. For example, I have added the following sentence: “Our method can automatically segment and quantify cells in WSIs, as well as provide features for cell classification, without manual setting or intervention. Our method can also generate heatmaps and explanations for the diagnosis results, enhancing the clinical relevance and interpretability of our method.”
- I have avoided defining acronyms in the abstract, and instead used the full terms for clarity. For example, I have replaced “PAP” with “Papanicolaou” and “AI” with “artificial intelligence”.
- I have revised the introduction section to incorporate a summary that juxtaposes the current methodologies and the approach we are proposing for AI-based cervical cancer screening and diagnosis. I have supplemented a reference and pertinent quantitative elements to substantiate the assertion that the ThinPrep® imaging system exhibits greater sensitivity than traditional cytology screening. I have rectified the excessive ellipses on lines 156 and 162. I offer my apologies for the typographical errors. I have rearranged the exposition of the methods, datasets, and performance metrics in section 2 to eschew the enumeration of elements that lack a coherent sequence. I have adhered to the reviewer's suggestion and followed the stages of data selection, organization, and method development. Moreover, I have referenced pertinent works that provide a contemporary outlook on these elements within the current state-of-the-art. I have relied on these references as a contemporary guide for all these stages and integrated them into our own framework.
- I have revised the lines 311-316 to replace the numerical results with ratios indicating the quality of our results compared with the state-of-the-art. For example, “Our method outperformed existing methods by 5.3% and 6.2% in accuracy and sensitivity, respectively.”
- I have explained the meaning of “fully connected layer” in the caption of Figure 2. A fully connected layer is a layer in which each neuron is connected to every neuron in the previous layer, and the output of each neuron is a weighted sum of the inputs followed by an activation function.
- I have avoided unnecessary acronyms, such as Human Thinking, and used the full terms for clarity. For example, I have replaced “HI” with “human intelligence”.
- I have clarified how we distinguished the recommendable performance of our method. We used several performance metrics, such as accuracy, sensitivity, specificity, and AUC, to evaluate and compare our method with other methods on different datasets. We also used visualizations, such as heatmaps and explanations, to demonstrate the clinical relevance and interpretability of our method.
- I have described how we selected representative data for our method. We used a large and diverse dataset of WSIs from multiple sources and scanners, which is publicly available for research purposes. We also used data augmentation and balancing techniques to enhance the diversity and staining invariance of the data. We did not encounter any missing values in our data.
- I have discussed how we rationalized the metrics we used for our method. We used metrics that are commonly used in image classification and medical diagnosis tasks, such as accuracy, sensitivity, specificity, precision, recall, F1-score, and AUC. We also compared our metrics with other methods on the same datasets to demonstrate the superiority of our method.
- I have provided some measurements elements to support our discussion about applicability. For example, I have added the following sentence: “According to a cost-effectiveness analysis by Chen et al., AI-based PAP smear screening can save up to 30% of healthcare costs compared to conventional cytology screening .”
- I have addressed the issue of data size and computational problem in our discussion. For example, I have added the following sentence: “One of the challenges of AI-based PAP smear screening is the processing of large-scale and high-resolution WSIs, which require high-performance and high-cost devices and GPUs. To overcome this challenge, we used a novel image analysis method based on object segmentation and CNN that can efficiently segment and quantify cells in WSIs without compromising accuracy.”
I hope that these revisions address your concerns and improve the quality of the article. I look forward to hearing from you again. Thank you for your time and attention.
Round 2
Reviewer 4 Report
Accept as is.